# Effects of Injection Molding Process Parameters on the Chemical Foaming Behavior of Polypropylene and Polystyrene

**DOI:** 10.3390/polym13142331

**Published:** 2021-07-16

**Authors:** Chen-Yuan Chung, Shyh-Shin Hwang, Shia-Chung Chen, Ming-Chien Lai

**Affiliations:** 1Department of Mechanical Engineering, National Central University, Taoyuan City 320317, Taiwan; benlai@moldex3d.com; 2Department of Mechanical Engineering, Chien Hsin University of Science and Technology, Taoyuan City 320312, Taiwan; stanhwang@uch.edu.tw; 3Department of Mechanical Engineering, Chung Yuan Christian University, Taoyuan City 320314, Taiwan; shiachun@cycu.edu.tw

**Keywords:** exothermic chemical foaming agent, foam injection molding, semi-crystalline material, amorphous material

## Abstract

In the present study, semi-crystalline polypropylene (PP) and amorphous polystyrene (PS) were adopted as matrix materials. After the exothermic foaming agent azodicarbonamide was added, injection molding was implemented to create samples. The mold flow analysis program Moldex3D was then applied to verify the short-shot results. Three process parameters were adopted, namely injection speed, melt temperature, and mold temperature; three levels were set for each factor in the one-factor-at-a-time experimental design. The macroscopic effects of the factors on the weight, specific weight, and expansion ratios of the samples were investigated to determine foaming efficiency, and their microscopic effects on cell density and diameter were examined using a scanning electron microscope. The process parameters for the exothermic foaming agent were optimized accordingly. Finally, the expansion ratios of the two matrix materials in the optimal process parameter settings were compared. After the experimental database was created, the foaming module of the chemical blowing agents was established by Moldex3D Company. The results indicated that semi-crystalline materials foamed less due to their crystallinity. PP exhibits the highest expansion ratio at low injection speed, a high melt temperature, and a low mold temperature, whereas PS exhibits the highest expansion ratio at high injection speed, a moderate melt temperature, and a low mold temperature.

## 1. Introduction

With the development of environmental awareness, an increasing number of products have been created with an emphasis on structural weight reduction. Polymeric foam manufacturing was first introduced in the 1960s. Beyer and Dahl [1] mixed compounds that can be thermally expanded in thermoplastic resinous material to facilitate chemical foaming in the material structure. In the early 1980s, researchers at Massachusetts Institute of Technology developed microcellular foaming [2]. Foam products developed using this technology absorb sound [3], have low thermal conductivity [4], and are resistant to shrinkage and warpage [5]; these products can be applied in packaging and insulation. Foam injection molding is achieved with the aid of foaming agents, either a physical blow agent (PBA) or a chemical blow agent (CBA), which can be dosed into the polymer. PBA foaming involves the mixing of plastic melt and supercritical fluid (SCF) in a injection barrel [6]. The solubility of SCF is between that of liquid and gas; however, its diffusion coefficient is 10–100 times that of liquid. Therefore, the SCF mass transfer rate is higher than that of liquid. Moreover, SCF exhibits viscosity close to that of gas. These two characteristics enable SCF to dissolve in plastic melt. The pressure drop caused by the injection of the mixture into the cavity causes thermodynamic instability in the gas and nucleation in the plastic melt with both CBA and PBA foaming. Subsequently, the foams expand through internal pressure, forming a product with holes in its internal structure, thereby achieving weight reduction. Chemical foaming [7], which has been in development for numerous years, has been applied in food insulation and packaging. Chemical foaming involves the use of chemical blowing agents (CBAs) to trigger chemical reactions at the decomposition temperature, forming gas and solid residue. According to the types of CBAs applied, chemical foaming can be exothermic, endothermic, or exo-endothermic balanced; exothermic blowing agents have the most favorable foaming efficiency [8] and are thus the most widely applied in industries. Chemical foaming surpasses physical foaming in economic and engineering efficiency. Solid residues can be used as nucleation centers to enhance cell structure in lightweight products. However, gases that are hazardous to humans are generated and cause combustion inside chemical foaming products in storage at high temperatures [9].

CBAs have been used in many foaming applications for some time. Hong et al. [10] studied the transport mechanism in polyurethane at room temperature and various CBA pressures. Most CBAs were hydro-chlorofluorocarbons (HCFCs). They also compared the diffusivity and solubility data of CBAs with those of chlorofluorocarbon 11 (CFC 11). They concluded that HCFCs can replace CFC as blowing agents. Kim et al. [11] studied the viscoelastic property of a saturated aliphatic polyester, poly(butylene adipate-*co*-succinate) (PBAS), cured by dicumyl peroxide (DCP). The effects of the additive content, foaming temperature, and curing agent content on the blowing ratio were also investigated. A closed-cell structure PBAS foam with a high blowing ratio (density of approximately 0.05 g/cm^3^) was obtained by adding 3 phr DCP. Reglero Ruiz et al. [12] used three endothermic CBAs (polyethylene-based compounds) and polypropylene to study expansion ratios and cellular morphologies. They observed that the cell size from a CBA based on citric acid was much smaller than that from CBA based on sodium bicarbonate due to the amount of the gas released. The expansion ratio varied between 1.4 at 0.5 MPa and 2 at 0.25 MPa.

Foams are generated in two processes: nucleation and cell growth. A greater number of nucleation sites results in more substantial foaming. Studies have explored factors that affect foaming. For example, in 1978, Villamizar and Han [13] reported that mold temperature affects foaming critically. In 1981, Han and Yoo [14] discovered that more severe short-shots lead to larger cell sizes. In 1984, Bhatti et al. [15] reported that higher melt temperature causes earlier chemical reactions in exothermic CBAs, producing gases and reducing product weight. In 1987, Colton and Suh [2] discovered that higher saturation pressure generates more number of bubbles. This phenomenon is believed to be influenced by injection speed; higher injection speed requires higher injection pressure, which causes a greater pressure drop when the melt mixture is injected into the cavity. Summarizing the aforementioned studies, Lee et al. [16] explored the approach required to achieve a uniform cell structure and a high expansion ratio. High-density polyethylene, which is high in viscosity, was applied to generate a fine-celled structure; talc was added to increase the number of foam nucleation sites, thereby further increasing the foaming performance. The results also reveal that pressure drop and gas content positively influence foaming results. Guo et al. [17] applied maleic anhydride grafted polypropylene (PP) and nanomaterials with an azodicarbonamide (AC) CBA added, revealing that viscosity critically affects foaming; excessively low material viscosity suppress cell nucleation. Lee [18] contended that increasing injection speed or screw revolutions per minute promotes shear force, which lowers the energy barrier for foaming and accelerates foam growth. Recently, a core-back foam technique was deployed to increase the weight reduction in the foam parts, especially in sport shoes applications. The required specific weight is approximately 0.2. Chu et al. [19] utilized a numerical simulation method to compare the simulation and experimental results in terms of the foaming temperature and the properties of the core-back foam injection process. Two viscosities of PPs were chosen for the crystallinity study. In their results, the PP with a low melt flow index had low crystallinity, a high crystalline rate, and a low crystallization temperature during cooling. The simulated and experimental results were consistent. Wu et al. [20] investigated the effects of process conditions (CBA dosage, shot size, mold temperature, injection speed, packing pressure, and core-back speed) on the weight reduction and tensile strength of a core-back chemical foaming process. Wu et al. [21] used the same technology to study the effect of core-back foaming on the weld-line strength. A special reticular structure was observed near the weld-line area. This reticular structure increased the weld-line strength. The aforementioned studies were referenced for assessing the feasibility of the experimental results in the present study.

Several experimental design strategies may be considered for improving injection molded product quality based on process parameter evaluation. Among the various experimental design strategies, the Taguchi method and response surface methodology have been widely used to determine the optimal process parameters for injection molding [22,23]. In the small factor change problem [24], because changing numerous factors is undesirable, some variation on a one-factor-at-a-time (OFAT) strategy would be appropriate for undertaking quality improvement with minimal changes to factor levels. In addition, a series of investigations [25,26] has demonstrated that OFAT has advantages over factorial experimental designs when the experimental error is small or when interactions among control factors are large.

Although physical foaming has been demonstrated to outperform chemical foaming, it requires expensive machines. Therefore, most industry owners are inclined to favor chemical foaming. The preparation of CBAs has improved significantly; both chemical reactions and foaming processes have improved considerably in efficiency. However, few investigations of CBA foaming in injection molding have been reported due to need for a shut-off nozzle on machines. This study examined the design of experiments on CBAs, hoping to discover optimal injection parameters to improve product foaming efficiency. The relationship between macroscopic effects (weight and specific weight) and microscopic effects (cell density and size) of a foaming product was explored. To the best of the authors’ knowledge, no study has adopted the OFAT experimental design to examine in further depth the effect of chemical foaming process conditions on expansion ratios. The experimental database in this study can serve as a guideline for the CBA foaming module of Moldex3D.

## 2. Experimental Method and Molding Simulation

### 2.1. Facility

The injection molding machine employed in this study was an Arburg Allrounder 420C (Arburg GmbH, Lossburg, Germany) equipped with Mucell capability and a screw diameter of 40 mm. The screw L/D (length to diameter) ratio of 24 is slightly longer than that of a conventional screw due to the extra plasticization. However, it has a shut-off nozzle that can be used in the CBA foaming process. The temperature of the mold was controlled using the BYCW-021410FS mold temperature controller created by the Byyoung International Company (New Taipei City, Taiwan). Product weight was measured using the LB-210S precision digital balance machine manufactured by LWL Germany Make (Göttingen, Germany). Product density was examined using the MH-300E electronic densimeter provided by MatsuHaku (Taichung City, Taiwan). The expansion ratio *ϕ* was calculated using Equation (1) [27]; a smaller $\rho_{f}$ indicates higher foaming efficiency and a higher expansion ratio. The goal of this study was to maximize the expansion ratio through the optimization of process parameters.
(1)$$\phi =\frac{\rho_{0}}{\rho_{f}}$$

$\rho_{f}$ = density after foaming; $\rho_{0}$ = density before foaming.

Figure 1 illustrates the location for the cutting of the specimen to remove a shape of length 1 cm and width 0.5 cm. The specimen was coated with 20-nm-thick platinum in a Cressington 208HR sputter coater equipped with a MTM-20 thickness controller (Ted Pella, Inc., Redding, CA, USA). A Tescan model Vega II scanning electron microscope (SEM; Tescan Orsay Holding, Brno, Czech Republic) was then used to profile the core region of the specimen at an accelerating voltage of 0.2–30 V and an image resolution of 3 nm at 30 kV. The open-source software program ImageJ, provided by the US National Institutes of Health, was then employed to process the image and calculate the sizes and numbers of cell, which were then applied in Equation (2) [28] to calculate cell density N_0_.
(2)$$N_{0}={\left(\frac{n}{A}\right)}^{\frac{3}{2}}$$

n = the number of cells in the micrograph; A = the area of the micrograph (µm^2^).

### 2.2. Material

To study the foam morphological discrepancy between semi-crystalline and amorphous polymers, PP and PS were adopted as the matrix materials. The PP was a Tairipro K1035 semi-crystalline material manufactured by the Formosa Chemicals & Fibre Corporation (Changhua County, Taiwan); its ideal melt temperature, mold temperature, and density were 200–290 °C, 30–50 °C, and 0.907 g/cm^3^, respectively. The general-purpose PS was a PG-33 amorphous material manufactured by the Chi Mei Corporation (Tainan City, Taiwan); its ideal melt temperature, mold temperature, and density were 185–205 °C, 40–70 °C, and 1.042 g/cm^3^, respectively. The apparent viscosity was measured using a capillary rheometer (CEAST SmartRHEO, Pianezza, Italy).

The CBA employed in this study was an AC exothermic foaming agent (AC-3000F), provided by Union Chemical Industry Company (Taipei City, Taiwan), which typically decomposes at the temperature of 200–206 °C [29], releasing 213–223 cm^3^/g of N_2_ through reactions. Because of the narrow range of decomposition temperature, this CBA is easy to control.

### 2.3. Molding Condition Setting

After the experimental database was created, the CBA foaming module was established by Moldex3D Company (Hsinchu County, Taiwan). In turn, the CBA foaming module of Moldex3D R16 was adopted for injection molding simulation, the result of which was compared with the results of the actual sample created with a short shot. Figure 2 depicts the sample and runner geometries. The material trade names of the PP and PS were defined in the model, and the following experimental process parameters were set: injection speed of 100 cm^3^/s, melt temperature of 210 °C, mold temperature of 60 °C, and CBA dosage of 1 wt%. The rate of formation of bubbles can be described using the cell nucleation model [30], where the threshold of bubble was set as default to 0.1 cm^−3^·sec^−1^. The bubble growth behavior model proposed by Han and Yoo was adopted [14], and the shot weight percentage was selected using the shot weight control for a weight reduction effect identical to that for the product. According to the experimental results, the weights of the solid parts in the PP and PS samples were 31.349 g and 39.99 g, respectively. The apparent viscosities of the PP and PS were measured at 210 °C (Figure 3). The apparent shear viscosity $\eta_{a}$ is defined as Equation (3) [31]. As indicated by Figure 3, the apparent viscosity of the PS is higher than that of the PP.
(3)$$\eta_{a}=\frac{\tau_{wa}}{{\dot{\gamma }}_{wa}}$$

$\tau_{wa}$ is the apparent shear stress at the wall; ${\dot{\gamma }}_{wa}$ is the apparent shear rate at the wall.

## 3. Experimental Design

Process parameters are key factors affecting the properties of injection molded parts. Process parameters include injection pressure, injection speed, melt temperature, mold temperature, packing pressure, packing time, and screw rotational speed. However, packing is not required in microcellular CBA foam injection molding. Melt temperature, mold temperature, and injection speed substantially affect the properties of foam molded parts [32,33,34] and were considered in this study. To clarify the extent of each factor’s influence, the OFAT experimental design was adopted. Each factor was divided into three levels. Table 1 and Table 2 provide the value of each factor at each level as well as the experiment design. Ten samples were generated from each experiment run for analysis of variance to determine the robustness of the manufacturing process. All products were created in short-shot molding for observation of the patterns of the unconfined expansion of foams. Because the foaming efficiency of blowing agent in samples created through full-shot molding may have been lowered because of the limitation caused by the mold walls, short-shot molding was adopted. The shot size in short-shot molding is two-thirds that in full-shot molding. The shot sizes were set as 45 and 60 cm^3^ for the PP and PS samples, respectively. See Table 3 for the other fixed parameters.

## 4. Results and Discussion

Foam morphology has been studied from several perspectives. The effects of process parameters on foam morphology were discussed by Xu [3]. They reported that amorphous materials (PS, PMMA, etc.) have a wider process window than semi-crystalline materials (PP, PE, etc.); therefore, they produce a more uniform cell structure. In another study, the skin layer, shear layer, and core region exist in the thickness of the injection-molded parts due to the characteristics of the injection molding process. The maximum shear rate (shear stress) occurred at 1/10 thickness under the skin layer of a sample [35].

### 4.1. Melt-Front Comparison of Simulated and Experimental Results

The model parameters used to compare the simulated results with the experimental results were those of the medium-level process (i.e., 100 cm^3^/s injection speed, 210 °C melt temperature, and 60 °C mold temperature). The melt front revealed a similarity between the simulated and experimental results in flows (Figure 4). The flow length was measured from the center of the box base (gating location) to the center of the short-shot side. The flow length on each side was quantified for comparison (Figure 5). The error in the flow lengths of the shorter sides (i.e., sides I and III) in the PP sample was 26.24%; those in the flow lengths of the longer sides (i.e., sides II and IV) was 11.51%. The error in the flow lengths of the shorter sides (i.e., side I and III) in the PS sample was 9.26%. Both the simulated and experimental results for the flow lengths of the longer sides of the PS sample (i.e., sides II and IV) reached the margin of 90 mm. The discrepancy of the flow length on sides I/III and II/IV was caused by the mold precision and gravity (the mold was positioned on the machine such that the longer side was vertical to the ground level). The actual wall thickness was not uniform on each side; however, uniform thickness was assumed in the simulation. Greater wall thickness resulted in longer flow length.

### 4.2. Effect of Injection Speed

According to Chen et al. [36], a higher shear rate causes an increased melt temperature due to shear heating, thus promoting cell nucleation and decreasing viscosity in PS melt. In turn, higher shear stress (shear rate) leads to favorable cell nucleation. As shown in Table 4 and Figure 6b, the weight and specific weight of the PS sample decreased significantly as the injection speed increased, whereas no significant deviation was noted in those of the PP samples among the three levels of injection speed. These PP foam results are in agreement with the results for the PET foam studied by Gómez-Gómez et al. [37]. According to their report, a higher injection speed resulted in larger cores [37]. A high injection speed causes high shear heating, which occurs under the skin layer, thus inducing a large region of core [35]. Among other reasons, PP is a semi-crystalline material, whereas PS is an amorphous material. PP exhibits higher specific heat than PS does; therefore, shear heating caused by an increase in the injection speed leads to an insignificant change in the PP temperature. Hence, it is insufficient to cause the blowing agent to decompose violently. Moreover, because PS exhibits higher viscosity (Figure 3) than PP does, PS receives higher shear stress than PP does at equal injection speed. As illustrated in Figure 7, at equal injection speed, the PS sample exhibited a higher expansion ratio than the PP sample did.

No significant macroscopic change was detected in the weight of the PP sample. As for at the microscopic level (Figure 8b,c), injection speed did not affect the cell density or cell size significantly (Table 5). Chen et al. [36] claimed that the cell density is significantly affected by shear stress nucleation resulting from the transformation of mechanical shear energy into surface energy. Our PS sample exhibited more favorable foaming at a higher injection speed; however, the cell morphology changed from closed to coalesced, as depicted in Figure 8f. Accordingly, although increasing injection speed promotes foam nucleation and growth, excessively rapid expansion causes open-cell structure, lowering the overall cell density (Table 5). Higher injection speed leads to higher expansion ratio. This is in consistent with findings by Chen et al. [36].

### 4.3. Effect of Melt Temperature

A solution saturation temperature exists for gas and plastic melt under a given pressure. The back pressure and compression zone of the screw ensured the single-phase mixture of gas and plastic melt. As shown in Table 6 and Figure 9, when the melt temperature was 200 °C, which was lower than the decomposition temperature of the exothermic foaming agent (200–206 °C), the weight and specific weight of the samples were maximal. A high melt temperature led to more violent exothermic foaming agent reactions, a greater amount of gas generated, and thus more gas participating in foaming, causing a decrease in the weight and specific weight of both matrix materials. As depicted in Figure 10, raising melt temperature caused an increase in the PP expansion ratio, which is consistent with the findings in other studies that higher melt temperature causes exothermic foaming agent to generate more gas [15] and that higher gas concentration leads to higher degree of foaming [16]. Raising the melt temperature increases foaming in PP, and this tendency was observed in the present experiment. The 220 °C melt temperature caused a change in the property of the PS sample because its processing temperature of 185–205 °C was exceeded. A higher melt temperature forced the gas out of the solution. In turn, fewer cells were observed in the cell morphology. Thus, this led to a slight increase in the specific weight of the PS sample.

As illustrated in Figure 11a and Table 7, no CBA decomposition was detected at the melt temperature of 200 °C; no cells were generated in the core region. As the melt temperature increased beyond 200 °C; however, a subsequent rise in the kinetic energy of the gas led to an increase in the foam growth rate. As shown in Figure 11, the cell size increased following the rise in the melt temperature.

### 4.4. Effect of Mold Temperature

According to Khoukhi and Tahat [38], a greater difference between PS temperature and the ambient temperature leads to lower PS density. In an injection molding experiment, ambient temperature is considered to equal mold temperature; higher mold temperature reduces the difference between PS temperature and ambient temperature, leading to greater PS density. This is consistent with the findings in the present study, for which higher mold temperature led to higher weight and specific weight in the PS sample (Table 8 and Figure 12). Figure 13 illustrates the changes in the expansion ratio as mold temperature changed. Under the short-shot condition, the mold temperatures for both the PP and PS samples were set as 50 °C to maximize their expansion ratios. According to Kawashima and Shimbo [39], higher mold temperature leads to larger cell size and lower cell density. This is because high temperature lowers viscosity, facilitating foam growth. As observed through the SEM (Table 9 and Figure 14), higher mold temperature led to larger cell size and lower cell density in the samples, consistent with the argument of Kawashima and Shimbo [39].

### 4.5. Optimal Process Parameter Settings

Table 10 lists the average expansion ratios of the PP and PS matrix materials in the exothermic foaming agent, which were determined through OFAT analysis of injection speed, melt temperature, and mold temperature. According to the experimental results, the effects of the parameters on the semi-crystalline and amorphous materials were consistent. To maximize the expansion ratios, a low injection speed, high melt temperature, and low mold temperature were set for the PP sample (80 cm^3^/s, 220 °C, and 50 °C, respectively). In contrast to the PP sample, a high injection speed, medium melt temperature, and low mold temperature were set for the PS sample (120 cm^3^/s, 210 °C, and 50 °C, respectively). This resulted in the average expansion ratios of 1.242 (±0.009) for the PP sample and 1.334 (±0.071) for the PS sample, respectively (Table 11).

## 5. Conclusions

Table 10 lists the average expansion ratios as obtained through the OFAT experimental design. Overall, the expansion ratio of the PS sample was higher than that of the PP sample. This was because PS is an amorphous material, and its specific volume is not considerably affected by temperature or pressure, thereby providing foams with sufficient space to expand. Furthermore, because PS exhibits a higher viscosity than PP does, PS is more easily affected by shear stress than PP. In summary, PS exhibits a higher expansion ratio and cell density than PP does.

The cell density data obtained in the present study were compared with those acquired by Guo et al. [17], revealing that all the cell density values fell within the 10^7^ cc^−1^ range. Guo et al. [17] applied both nanocomposits and an exothermic CBA (AC-3000F) to increase foaming efficiency. In the present study, only pure PP was used. Because the nonpolar molecules of PP are not grafted, its miscibility with other polymers was poor. Therefore, at some of the manufacturing parameter settings, the cell density in the present study was lower than that in the findings by Guo et al. [17].

In the OFAT experimental design, optimal parameters were determined for the selected matrix materials, and higher expansion ratios were obtained than those in the experimental planning (Table 10). The expansion ratios of PP and PS increased by 6.25% and 13.15%, respectively, verifying that OFAT parameter optimization effectively improved the quality of the foaming products.

This study employed the short-shot approach to examine the effect of various manufacturing parameters on the foaming phenomenon. Under the short-shot condition, the melt front was not limited by the mold walls. The simulated and experimental flow lengths were compared to verify the accuracy of the simulations, and the comparison revealed the error between the simulated and experimental results to be ≤30%. In future studies focusing on the full-shot condition, simulated data on volumetric shrinkage, weight, and cell density can be compared with experimental data. The use of the OFAT experimental design is generally discouraged by experts in experimental design strategies and quality improvement [40]. However, substantial support exists for the proposal that OFAT can be more effective than orthogonal arrays when the pure experimental error is low or interactions between parameter settings are strong [25,26]. Although the OFAT approach reacts more quickly to data, future studies may adopt Taguchi’s orthogonal arrays to systematically and efficiently adjust control factors or may employ a response surface methodology to examine interactions among control factors, thus verifying the reliability of the present study.

## Figures and Tables

**Figure 1 polymers-13-02331-f001:**
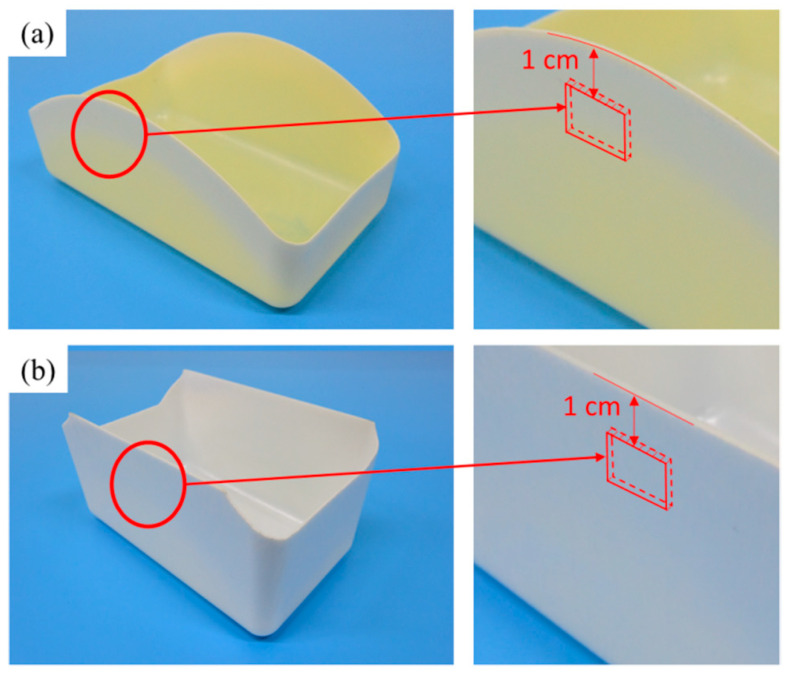
(**a**) Cutout location of the PP specimen for scanning electron microscope (SEM) analysis; (**b**) cutout location of the PS specimen for SEM analysis.

**Figure 2 polymers-13-02331-f002:**
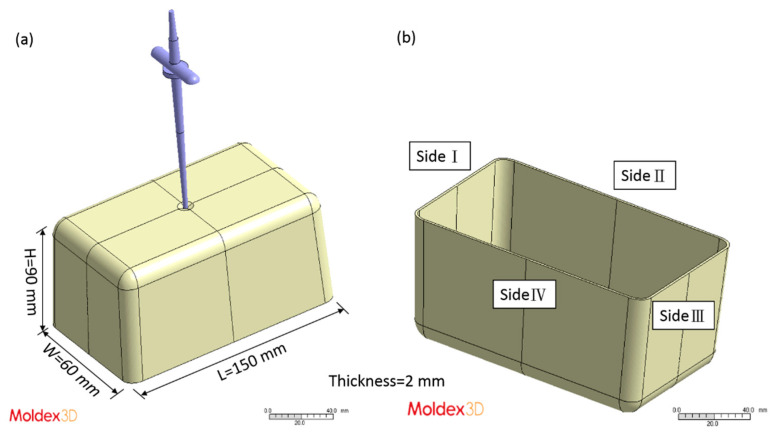
(**a**) Model including sample and runner; (**b**) naming of the sample sides.

**Figure 3 polymers-13-02331-f003:**
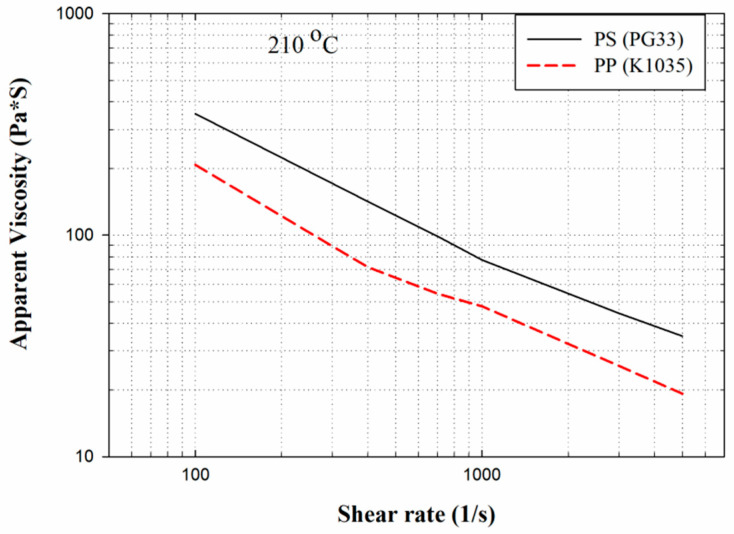
Apparent viscosities of PP and PS at 210 °C.

**Figure 4 polymers-13-02331-f004:**
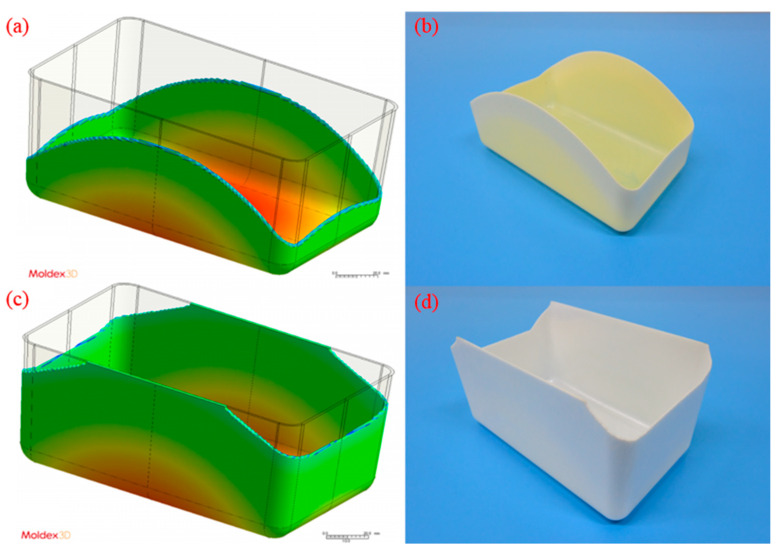
Simulated and experimental results using medium-level process parameters (i.e., 100 cm^3^/s injection speed, 210 °C melt temperature, and 60 °C mold temperature): (**a**) simulated result for the PP sample, (**b**) actual experimental result for the PP sample, (**c**) simulated result for the PS sample, (**d**) actual experimental result for the PS sample.

**Figure 5 polymers-13-02331-f005:**
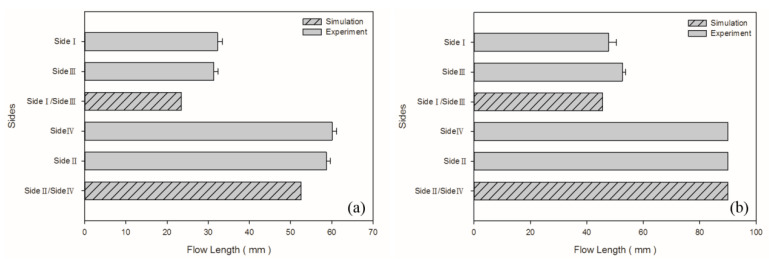
Experimental and simulated flow lengths for medium-level parameters in the process: (**a**) PP matrix and AC CBA; (**b**) PS matrix and AC CBA.

**Figure 6 polymers-13-02331-f006:**
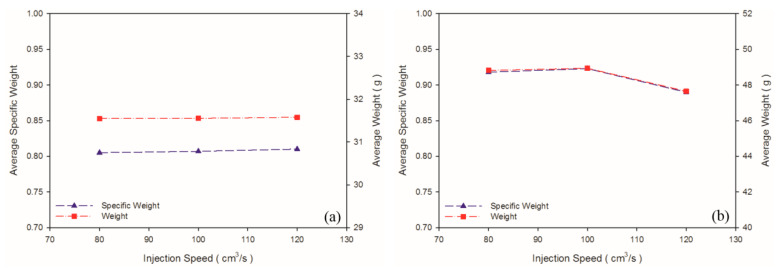
Weights and specific weights at different injection speeds: (**a**) PP matrix and AC CBA; (**b**) PS matrix and AC CBA.

**Figure 7 polymers-13-02331-f007:**
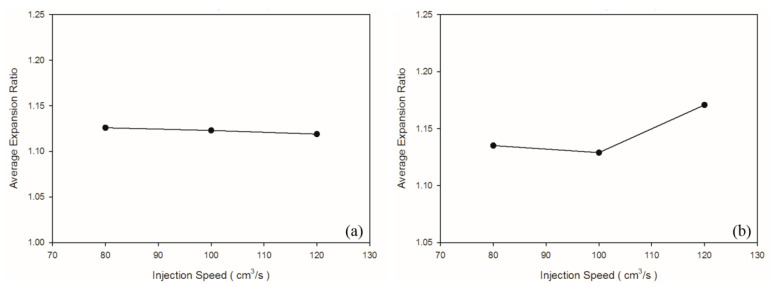
Expansion ratios at various injection speeds: (**a**) PP matrix and AC CBA; (**b**) PS matrix and AC CBA.

**Figure 8 polymers-13-02331-f008:**
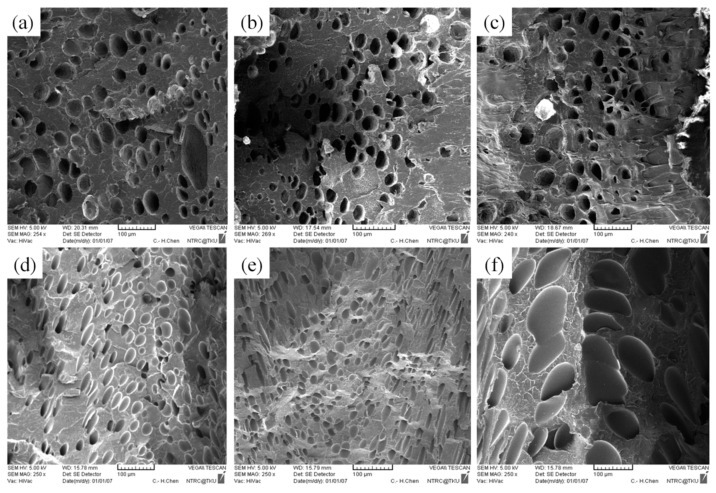
SEM micrographs of PP processed with different injection speeds: (**a**) 80 cm^3^/s, (**b**) 100 cm^3^/s, (**c**) 120 cm^3^/s; SEM micrographs of PS processed with different injection speeds: (**d**) 80 cm^3^/s, (**e**) 100 cm^3^/s, (**f**) 120 cm^3^/s.

**Figure 9 polymers-13-02331-f009:**
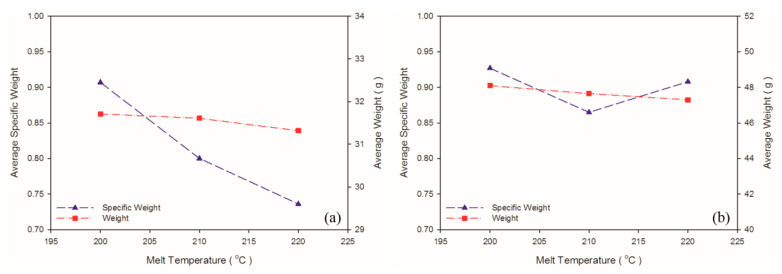
Weight and specific weight at different melt temperatures: (**a**) PP matrix and AC CBA; (**b**) PS matrix and AC CBA.

**Figure 10 polymers-13-02331-f010:**
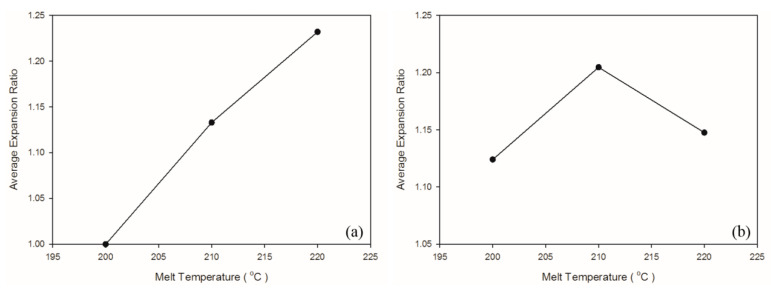
Expansion ratio at various melt temperatures: (**a**) PP matrix and AC CBA; (**b**) PS matrix and AC CBA.

**Figure 11 polymers-13-02331-f011:**
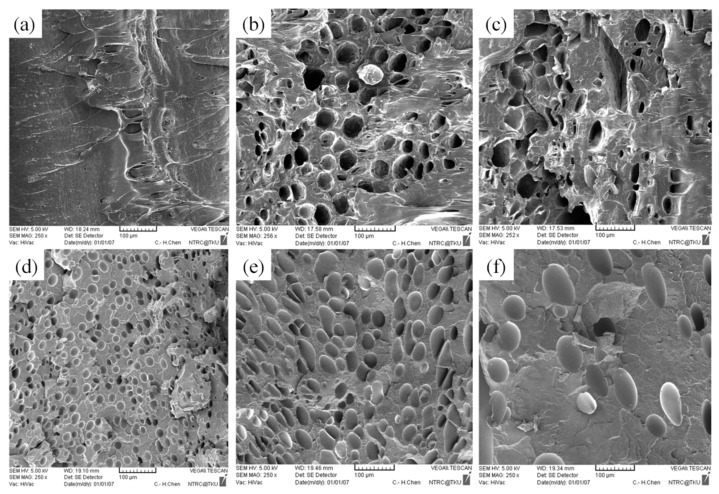
SEM micrographs of PP for different melt temperatures: (**a**) 200 °C, (**b**) 210 °C, (**c**) 220 °C; SEM micrographs of PS for different melt temperatures: (**d**) 200 °C, (**e**) 210 °C, (**f**) 220 °C.

**Figure 12 polymers-13-02331-f012:**
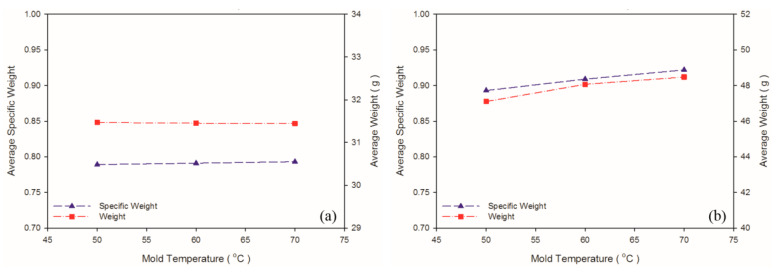
Weight and specific weight at different mold temperatures: (**a**) PP matrix and AC CBA; (**b**) PS matrix and AC CBA.

**Figure 13 polymers-13-02331-f013:**
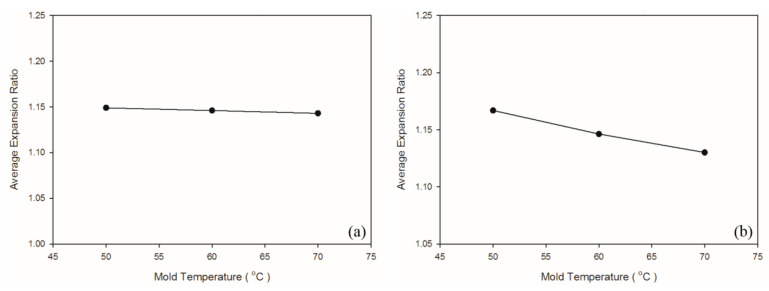
Expansion ratios at various mold temperatures: (**a**) PP matrix and AC CBA; (**b**) PS matrix and AC CBA.

**Figure 14 polymers-13-02331-f014:**
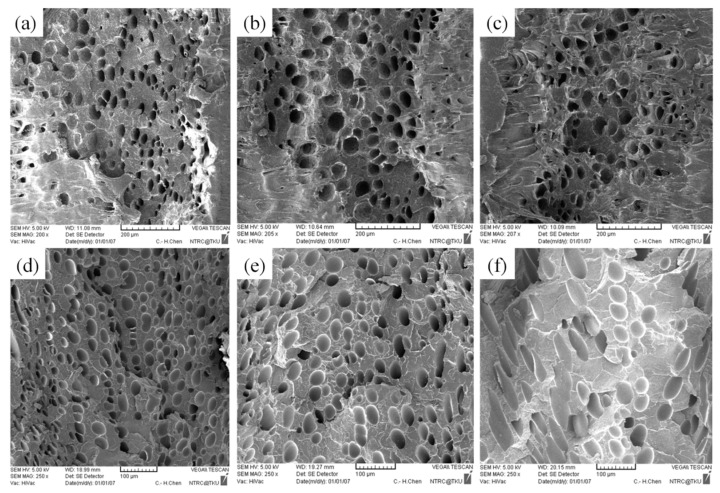
SEM micrographs of PP for different mold temperatures: (**a**) 50 °C, (**b**) 60 °C, (**c**) 70 °C; SEM micrographs of PS for different mold temperatures: (**d**) 50 °C, (**e**) 60 °C, (**f**) 70 °C.

**Table 1 polymers-13-02331-t001:** Factor levels for the experiment.

Factors	Low (Level 1)	Medium (Level 2)	High (Level 3)
Injection Speed (cm^3^/s)	80	100	120
Melt Temperature (°C)	200	210	220
Mold Temperature (°C)	50	60	70

**Table 2 polymers-13-02331-t002:** One-factor-at-a-time experimental design.

Run	Injection Speed	Melt Temperature	Mold Temperature
1	Level 1	Level 2	Level 2
2	Level 2	Level 2	Level 2
3	Level 3	Level 2	Level 2
4	Level 2	Level 1	Level 2
5	Level 2	Level 2	Level 2
6	Level 2	Level 3	Level 2
7	Level 2	Level 2	Level 1
8	Level 2	Level 2	Level 2
9	Level 2	Level 2	Level 3

**Table 3 polymers-13-02331-t003:** Other fixed parameters.

Setting	
Shot Size (cm^3^)	PP = 45; PS = 60
Injection Pressure (bar)	1500
V/P Point (cm^3^)	5
Packing Pressure (bar/s)	400/0.2 → 200/0.2 → 30/0.2
Tangent Speed of Screw (m/min)	25
Cooling Time (s)	35
Dispersed Oil (wt%)	1.0
CBA (wt%)	1.0

**Table 4 polymers-13-02331-t004:** Macroscopic effects of injection speed on weight, specific weight, and expansion ratio.

**Polypropylene**			
Injection Speed (cm^3^/s)	80	100	120
Weight (g)	31.55 (±0.048)	31.56 (±0.008)	31.58 (±0.007)
Specific Weight	0.805 (±0.002)	0.807 (±0.003)	0.810 (±0.003)
Expansion Ratio	1.126 (±0.002)	1.123 (±0.003)	1.119 (±0.004)
**Polystyrene**			
Injection Speed (cm^3^/s)	80	100	120
Weight (g)	48.82 (±0.586)	48.94 (±0.413)	47.65 (±1.418)
Specific Weight	0.918 (±0.020)	0.923 (±0.019)	0.890 (±0.013)
Expansion Ratio	1.133 (±0.024)	1.127 (±0.022)	1.169 (±0.017)

**Table 5 polymers-13-02331-t005:** Microscopic effects of injection speed on the PP and PS cells as observed through the SEM.

**Polypropylene**			
Injection Speed (cm^3^/s)	80	100	120
Cell Density (1/cc)	$5.893\times 10^{6}$	$1.132\times 10^{7}$	$8.609\times 10^{6}$
Cell Diameter (µm)	42.434	32.887	40.292
**Polystyrene**			
Injection Speed (cm^3^/s)	80	100	120
Cell Density (1/cc)	$1.801\times 10^{7}$	$4.055\times 10^{7}$	$2.098\times 10^{6}$
Cell Diameter (µm)	39.041	32.337	95.469

**Table 6 polymers-13-02331-t006:** Macroscopic effects of melt temperature on weight, specific weight, and expansion ratio.

**Polypropylene**			
Melt Temperature (°C)	200	210	220
Weight (g)	31.71 (±0.027)	31.62 (±0.042)	31.33 (±0.053)
Specific Weight	0.907 (±0.001)	0.800 (±0.002)	0.736 (±0.003)
Expansion Ratio	1.000 (±0.001)	1.134 (±0.032)	1.232 (±0.006)
**Polystyrene**			
Melt Temperature (°C)	200	210	220
Weight (g)	48.10 (±0.936)	47.65 (±1.404)	47.30 (±0.917)
Specific Weight	0.927 (±0.018)	0.865 (±0.019)	0.908 (±0.009)
Expansion Ratio	1.123 (±0.021)	1.202 (±0.027)	1.145 (±0.012)

**Table 7 polymers-13-02331-t007:** Microscopic effects of melt temperature on the PP and PS cells as observed through the SEM.

**Polypropylene**			
Melt Temperature (°C)	200	210	220
Cell Density (1/cc)	N/A	$7.759\times 10^{6}$	$7.242\times 10^{6}$
Cell Diameter (µm)	N/A	41.980	47.927
**Polystyrene**			
Melt Temperature (°C)	200	210	220
Cell Density (1/cc)	$3.918\times 10^{7}$	$1.756\times 10^{7}$	$7.271\times 10^{5}$
Cell Diameter (µm)	27.057	57.444	130.113

**Table 8 polymers-13-02331-t008:** Macroscopic effect of mold temperature on weight, specific weight, and expansion ratio.

**Polypropylene**			
Mold Temperature (°C)	50	60	70
Weight (g)	31.47 (±0.044)	31.46 (±0.020)	31.45 (±0.020)
Specific Weight	0.789 (±0.008)	0.791 (±0.005)	0.793 (±0.008)
Expansion Ratio	1.150 (±0.012)	1.146 (±0.007)	1.143 (±0.012)
**Polystyrene**			
Mold Temperature (°C)	50	60	70
Weight (g)	47.10 (±0.844)	48.07 (±0.838)	48.48 (±0.310)
Specific Weight	0.893 (±0.011)	0.909 (±0.025)	0.922 (±0.004)
Expansion Ratio	1.165 (±0.015)	1.145 (±0.031)	1.128 (±0.005)

**Table 9 polymers-13-02331-t009:** Microscopic effects of mold temperature on the PP and PS cells as observed through the SEM.

**Polypropylene**			
Mold Temperature (°C)	50	60	70
Cell Density (1/cc)	$1.075\times 10^{7}$	$7.066\times 10^{6}$	$5.696\times 10^{6}$
Cell Diameter (µm)	40.260	46.089	74.692
**Polystyrene**			
Mold Temperature (°C)	50	60	70
Cell Density (1/cc)	$2.798\times 10^{7}$	$1.337\times 10^{7}$	$5.028\times 10^{6}$
Cell Diameter (µm)	33.167	39.585	51.171

**Table 10 polymers-13-02331-t010:** Average expansion ratios according to OFAT parameter settings.

**Average expansion ratio of polypropylene**				
	**Levels**	**Low**	**Medium**	**High**
**Parameters**				
Injection speed (cm^3^/s)		1.126	1.123	1.119
Melt temperature (°C)		1.000	1.134	1.232
Mold temperature (°C)		1.150	1.146	1.143
**Average expansion ratio of polystyrene**				
	**Levels**	**Low**	**Medium**	**High**
**Parameters**				
Injection speed (cm^3^/s)		1.133	1.127	1.169
Melt temperature (°C)		1.123	1.202	1.145
Mold temperature (°C)		1.165	1.145	1.128

**Table 11 polymers-13-02331-t011:** Expansion ratios with the optimal process parameter settings.

Materials	Parameters		Expansion Ratio
PP	Injection Speed	80 cm^3^/s	1.242 (±0.009)
	Melt Temperature	220 °C	
	Mold Temperature	50 °C	
PS	Injection Speed	120 cm^3^/s	1.334 (±0.071)
	Melt Temperature	210 °C	
	Mold Temperature	50 °C	

## Data Availability

The data presented in this study are available on request from the corresponding author.

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
