# Peer review of "Effects of Injection Molding Process Parameters on the Chemical Foaming Behavior of Polypropylene and Polystyrene"

_polymers, 2021, doi:10.3390/polym13142331_

Round 1

Reviewer 1 Report

Thank you for the revisions.  For the revised paper,

1) There is a lot of extra information in the Introduction.  Please focus on the chemical foamed parts.  Please reduce the amount of information on physical blowing agents, microcellular foam molding, and core back foaming.

2) What is the L/D and overall design of the injection molding screw?  Specifically, did you use the screw designed for microcellular foaming?

3) Please remove the titles from the tops of the graphs in Figures 3, 5, and 6.  Incorporate the critical information into the figures titles. 

Author Response

Dear Reviewer,

Please download the attachment.

Sincerely yours,

Dr. Chen-Yuan Chung

Reviewer 2 Report

In the paper, an effect of injection molding parameters on the chemical foaming of polypropylene and polystyrene has been studied.

In general, the paper is interesting, relatively well written, however, of an engineering value.

It is not clear why the authors chose this optimization method. It's worth justifying it. The literature review in this area is very limited. There is a lot of papers on the optimization of injection molding, e.g.

- Wilczyński, K.; Narowski, P.A Strategy for Problem Solving of Filling Imbalance in Geometrically Balanced Injection Molds Polymers 2020, 12, 805; doi:10.3390/polym12040805

- Fernandes, C.; Pontes, A.J.; Viana, J.C.; Gaspar-Cunha, A. Modeling and Optimization of the Injection-Molding Process: A Review. Adv. Polym. Technol. 2018, 37, 429–449, [10.1002/adv.21683].

- Fernandes, C.; Pontes, A.J.; Viana, J.C.; Gaspar-Cunha, A. Using Multi-objective Evolutionary Algorithms for Optimization of the Cooling System in Polymer Injection Molding. Int. Polym. Proc. 2012, 27, 213–223, doi:10.3139/217.2511.

- Gaspar-Cunha, A.; Covas, J.A. Optimization in Polymer Processing; Nova Science Publishers: New York, USA, 2011; ISBN 978-1-61122-818-2.

- Fernandes, C.; Pontes, A.J.; Viana, J.C.; Gaspar-Cunha, A. Using Multiobjective Evolutionary Algorithms in the Optimization of Operating Conditions of Polymer Injection Molding. Polym. Eng. Sci. 2010, 50, 1667–1678, doi:10.1002/pen.21652.

The authors correctly write that “Use of OFAT experimental design is generally discouraged by experts in experimental design strategies and quality improvement”. The question is why they used this method ?

The authors performed some simulations, however, they did not describe it at all. In particular, simulation parameters should be given in terms of material and operation, including the initial nucleation parameters which are crucial for simulation. And, the foaming simulation module should be shortly described.

The authors write about viscosity (higher, lower), but there are no viscosity charts.

The authors write: “This resulted in the average expansion ratios of 1.242(±0.009) for the PP sample and 1.334(±0.071) for the PS sample, respectively (Table 11)”. What way were these values obtained ?

The authors use “injection speed” and “screw speed”, however the dimensions are different. May be, “injection rate” would be better.

Finally, this is an engineering research, and the authors has solved an engineering problem. They should expand the scientific elements of their studies,

Author Response

(The authors gave the same response as above.)

Round 2

Reviewer 2 Report

Thank you for revising the paper.